# Predicting Risk Factors of Lower Extremity Injuries in Elite Women’s Football: Systematic Review and Meta-Analysis

**DOI:** 10.3390/sports11090187

**Published:** 2023-09-20

**Authors:** Feim Gashi, Tine Kovacic, Arbnore Ibrahimaj Gashi, Arben Boshnjaku, Ismet Shalaj

**Affiliations:** 1Physiotherapy Program, Faculty of Medicine, Alma Mater Europaea—ECM, 2000 Maribor, Slovenia; feim.gashi@almamater.si; 2Physiotherapy Department, Faculty for Health Science, University of Ljubljana, 1000 Ljubljana, Slovenia; tine.kovacic@zf.uni-lj.si; 3Physiotherapy Program, Faculty of Medicine, University “Hasan Prishtina”, 10000 Pristina, Kosovo; arbnore.gashi@uni-pr.edu; 4Physiotherapy Program, Faculty of Medicine, University “Fehmi Agani”, 50000 Gjakova, Kosovo; arben.boshnjaku@uni-gjk.org; 5Physiotherapy Department, Faculty of Medical Sciences, Alma Mater Europaea Campus College Rezonanca, 10000 Prishtina, Kosovo

**Keywords:** lower extremity injuries, women, elite, football, players, risk factor

## Abstract

This study identified and analyzed the risk factors of lower extremity injuries (LEI) in elite women football players to improve career and health outcomes. To address this aim, a systematic review and meta-analysis methodology was used. In total, four relevant research articles were identified through database searching and screening using the PRISMA flow diagram. From these articles, eight predictors were identified that influence the risk of LEI among elite women football players: higher body mass index (OR 1.51, 95% CI); previous knee injury (OR 3.57, 95% CI); low normalized knee separation (≤10th percentile) (RR 1.92, 95% CI); all previous injury (previous ACL tear: OR 5.24, 95% CI; ankle sprain: 1.39, 95% CI; knee sprain: 1.50, 95% CI); and previous injury in the lower body (OR 2.97, 95% CI). Meanwhile, lower knee valgus angle in a drop-jump landing (OR 0.64, 95% CI) was found to decrease the risk of LEI among elite women football players.

## 1. Introduction

Injuries sustained by athletes are a long-standing and significant issue in virtually every form of outdoor sports. The impacts of such injuries on both individual and organizational levels are significant. Such player injuries not only lead to significant financial damages to their respective teams but also pose major ethical and regulatory concerns to professional sports leagues at large [1]. Meanwhile, severe sports injuries can lead to early retirement and long-term health issues in players, as injury rates range from 62%, 38.17%, and 60.83% to 66%, which can result in major negative outcomes related to the physical, mental, and overall well-being of players [2]. Therefore, reducing the prevalence and mitigating the impacts of sport-related injuries is essential for protecting the interests of both players and sports organizations.

As a complex contact sport, football is associated with a relatively high risk of injuries. According to Forsythe et al. [3], for every 1000 h of exposure, professional football players sustain between 4 and 35 injuries. The most common types of injuries sustained by football players during the game include adductor strains (7.6%), ankle sprains (8.5%), and hamstring strains (12.3%) [3], thus suggesting that the majority of injuries are lower extremity injuries (LEI), or injuries affecting parts of the lower extremity of the body [4]. Thus, it can be stated that LEI are the most prevalent type of injuries in football. While such injuries affect both male and female football players, contemporary research has proven that the risk factors and impacts of these injuries differ between these gender groups [5]. As such, specific research is necessary to identify the risk factors causing LEI in football players of each gender. Moreover, it has also been found that the recovery time and negative career and health impacts of injuries are particularly high in female football players [6]. Therefore, it is essential to understand the factors that cause a risk of LEI in female football players, to eliminate those risk factors, and minimize the risk of LEI in female football players.

A considerable body of research has already been conducted to identify and analyse the risk factors of LEI among female football players. However, most of these studies focus either on specific demographics of female football players or on specific types of LEI. Therefore, further secondary research is necessary to compare the findings of the relevant existing studies to identify the most prominent risk factors of LEI in female football players. Moreover, such a secondary analysis can also lead to potentially novel findings by providing a deeper understanding of how the identified risk factors influence the risk of LEI among female football players.

As mentioned earlier, injuries have profound effects on the career, health and overall well-being of female football players. Apart from direct negative impacts, football-related injuries have also been observed to cause some important long-term negative outcomes in women football players. Such outcomes include kinesiophobia or the fear of movement, higher anxiety, and depression due to career loss and competition anxiety [7]. On a physical level, LEI related to sports can result in severe health issues, such as shin splints, stress fractures, and tendonitis, which may significantly deteriorate the health of the affected individuals over time. However, numerous researchers [8,9,10] have explored the risk factors of LEI among female football players using different methods and sample demographics. By combining and comparing the findings of these studies through a systematic literature review, the most prominent risk factors of LEI in female football players can be identified, along with their interactions with the risk of LEI. These risk factors can then be modified to drastically reduce the risk of LEI in elite women’s football [8]. Thus, it can be stated that the findings of this systematic literature review study can be highly valuable for improving the career and health outcomes of elite women football players.

This study aims to identify and analyse the risk factors of lower extremity injuries in elite women football players.

## 2. Materials and Methods

The study was conducted using the combined method of a systematic review and meta-analysis. This research method is widely used to explore a research problem using data collected by previous researchers and their postulations [11]. In this section, the methods and criteria used to collect, extract and analyse data in this systematic review and meta-analysis-based study are described.

### 2.1. Eligibility Criteria

To select the studies with relevant sample profiles for the meta-analysis, the PICO (Population, Intervention, Control, and Outcomes) framework was used. Using this framework, the preliminary eligibility criteria for the samples of the studies to be included in this systematic review are defined below. Population: Elite women football players; Intervention: Risk factors of injury; Control: Elite women football players without lower extremity injury; Outcome: Lower extremity injury. The inclusion and exclusion criteria used to screen academic research articles identified through database searching for this study are listed in Table 1.

### 2.2. Search Strategy and Study Selection

The search strategy used in this study was specifically developed based on the research objectives of this study to identify the most relevant articles for the systematic review. The following databases were searched to identify potentially relevant research articles: MEDLINE, EMBASE, Cochrane, PubMed, Google Scholar, Web of Science, SCOPUS, and CINAHL. Collectively, these databases provide access to a vast body of high-quality research related to the topic of interest. As demonstrated by Snyder et al. [12], the keywords used to search the databases were extracted from the research questions and objectives of the present study. Hence, the keywords “lower extremity injuries”, “women”, “female”, “elite”, “soccer”, “football”, “players”, and “risk factor” were used to search the database. The risk factors of lower extremity injuries in women elite football players were the subject of search queries developed using these keywords. The keywords were joined to form the search queries using the Boolean operators, “AND”, “OR”, and “NOT”.

The PRISMA (Preferred Reporting Items for Systematic Reviews and Meta-Analyses) framework was applied to screen the search results and identify the most suitable research articles for this systematic review. In systematic literature reviews, the framework offers a methodical way to screen and eliminate irrelevant search results through a multi-step procedure [13]. As such, two consecutive screening techniques were used in this study to identify and eliminate irrelevant studies. In the first step, the titles and abstracts of the articles identified through database searching were screened for relevant keywords and information. The articles containing such keywords in their abstracts and titles were retained, while the rest were discarded [14]. After that, the text of the retained articles was similarly searched to determine whether the articles contained relevant information for this study. The articles found to be relevant to this review were included in the data analysis, and the rest were discarded.

### 2.3. Title/Abstract Screening

After searching the databases, two authors screened the abstracts and titles based on the inclusion and exclusion criteria. A manual approach was used to find relevant keywords through reading titles and abstracts. Afterwards, a third author went through the screening process again to ensure minimal bias. With the help of this screening method, the researcher was able to quickly limit viable data sources from a large body of evidence [14].

### 2.4. Full-Text Screening

After the initial screening of titles and abstracts, a full screening was conducted. The author used a manual approach for reading the selected studies manually and studies relevant to the inclusion and exclusion criteria were selected. After the initial screening of titles and abstracts, a total of 15 articles were selected, from which 11 were excluded. Out of these, six took consent after the study, four focused more on systemic injuries rather than lower extremities, and one had paid access only. Four studies passed the inclusion criteria and were selected for data extraction. The whole screening procedure involved three authors for minimal selection bias.

### 2.5. Data Extraction

Relevant data were extracted from the selected research articles using a systematic data extraction method. A data extraction table (Table 2) was used for data extraction and developed based on the data extraction table used by Ahn et al. [11] in their study. However, the table was customized to collect specific data related to this study. As a result, Table 2 was designed to collect the most relevant data to understand the risk factors of LEI in elite women football players.

### 2.6. Risk of Bias Assessment and Quality Assessment

The CASP (Critical Appraisal Skills Programme) checklists were used to assess the calibre of the contained literature. These checklists make use of a series of closed-ended questions to evaluate the calibre of various types of studies regarding their methodology, data collection methods, and research designs [15]. Different CASP checklists apply to different types of studies, including randomized controlled trials (RCT), cross-sectional studies, and qualitative studies. Based on the number of CASP criteria satisfied by each of the included studies, the studies were attributed a score out of 10. This led to an objective and accurate assessment of the quality of the included studies.

### 2.7. Statistical Analysis

As mentioned in Table 2, the odds ratio (OR) or risk ratio (RR) of each risk factor was extracted from the included articles. The OR and RR values provide an accurate understanding of the statistical significance of each of the risk factors, while also enabling the relative prioritization of the risk factors identified in this study [16].

## 3. Results

### 3.1. Descriptive Characteristics of the Studies

As depicted in Figure 1, four research articles were included in this study through a systematic screening of database search results. In total, these studies recruited 895 elite female football players. The prevalence of LEI in the studies varied significantly (from 38.17% to 66%). Among the four studies, two were prospective cohort studies: one was a cross-sectional cohort study, and one was a longitudinal cohort study. Finally, the average CASP assessment score of the included studies was 9.25 (between 9 and 10). Therefore, it can be stated that a set of high-quality data was included in the present study.

### 3.2. Findings

Table 3 is the data extraction table that shows the relevant extracted data and the findings of this study. Meanwhile, Table 4 shows the statistical findings of the included studies.

From Table 3 and Table 4, it can be observed that a total of six factors were identified as influencing the risk of LEI among women football players. These factors are higher body mass index (BMI) (OR 1.51, 95% CI); lower knee valgus angle in a drop-jump landing (OR 0.64, 95% CI); previous knee injury (OR 3.57, 95% CI); low normalized knee separation (≤10th percentile) (RR 1.92, 95% CI); previous injury (anterior cruciate ligament rupture: OR 5.24, 95% CI; ankle sprain: 1.39, 95% CI; knee sprain: 1.50, 95% CI); and previous injury in the lower body (OR 2.97, 95% CI). Different characteristics were studied differently in all four studies, as BMI was identified as a risk factor in only one study, while two studies identified previous injuries as a high risk. This resulted in heterogeneity between the characteristics of the selected studies.

Based on the influence of the factors on the risk of LEI in elite women football players and the nature of the factors, the identified risk factors are categorized into different groups in Table 5. From Table 5, it can be observed that one of the six factors (lower knee valgus angle in a drop-jump landing) identified in this study reduces the risk of LEI among elite women football players, whereas the rest effectively increase this risk.

## 4. Discussion

In this section, the findings described in the previous section are critically discussed using existing literature to formulate a cohesive narrative that addresses the objectives of the present study. To better understand the influence of each identified factor, the factors are grouped according to the categories mentioned in Table 5.

### 4.1. Injury Rates

As observed in Table 3, the rates of LEI among the participants significantly varied from one study to another, with the lowest injury rate being 38.17% and the highest rate being 66%. All research articles included in this systematic review study investigated the risk factors of all types of lower extremity injuries. Notably, although Faude et al. [18] aimed to explore the risk factors of all types of injuries in elite women football players, the data collected in this study were related only to LEI. Meanwhile, 19. Hägglund et al. [19] recruited both male and female elite football players in their study. However, data related to only female elite football players were extracted from the study by Hagglund [19] in this systematic review. One of the studies also included high school football players, where the participants were teenagers, and age could be linked as a factor influencing the development of LEI. However, since this baseline characteristic significantly varied and age was not reported in all studies, it could not be included as a sufficient predictor.

### 4.2. Predictors of LEI

As depicted in Table 5, a total of five themes were identified in relation to factors that increase the risk of LEI in elite women football players. These factors are High BMI; Low normalized knee separation (≤10th percentile); previous knee injury; previous injury in the lower body; and previous injury. Among these themes, two (High BMI; low normalized knee separation) are related to the physical characteristics of the players, whereas the other three (previous knee injury; previous injury in the lower body; and previous injury) are related to the history of injuries in the players. In addition to the risk factors, one factor that reduces the risk of LEI among women football players was identified. This factor is the Lower knee valgus angle in a drop-jump landing. Notably, this factor is related to the practices of the players. Based on the grouping of the identified themes, the following major predictors of risk factors of LEI in elite women football players were identified.

### 4.3. Previous Injury

Three of the four studies reviewed in this study implied that previous injury increases the risk of LEI in elite women football players. While Nilstad et al. [17] found that previous knee injury increases the risk of LEI in the population of interest, Faude and colleagues [18] found that three specific types of injuries increase the risk of LEI. These types of injury are anterior cruciate ligament rupture, ankle sprain, and knee sprain. However, according to Hagglund et al. [19], any kind of injury in the lower body in the past can effectively increase the risk of LEI in elite women football players. Notably, although the three studies focused on different types of injuries, the relative risk (or relative odds) of elite women football players sustaining LEI was found to be significant in each of these studies. Thus, it can be concluded that, regardless of the nature of the injury, any kind of LEI increases the risk of future injuries in elite women’s football players. Although the cause behind this correlation has not been satisfactorily described, other researchers have also found that previous injuries increase the risk of future injuries in elite women football players regardless of the location of the injury [7]. This risk factor can be partially explained based on the findings of Hagglund and colleagues [19]. The researchers found that athletes who have experienced injuries in the past are likely to repeat the same movements or components along the kinematic chain that led to the previous injuries, thereby increasing the risk of future injuries.

### 4.4. Knee Position and Movement

Nilstad and colleagues, as well as O’Kane and colleagues [9,19], found lower knee valgus angle in a drop-jump landing, and low normalized knee separation respectively to be associated with the risk of LEI in elite women football players. While lower knee valgus angle in a drop-jump landing was found to decrease the risk of LEI in the target population, low normalized knee separation was found to increase the risk of LEI in the same population group. These predictors can also be explained based on the findings of Hagglund at al. [19]. Both factors are components of the kinematic chain involved in the jumps and landings of elite women football players. Therefore, changes in these factors influence the probability of LEI injuries in such players.

### 4.5. BMI

Nilstad and colleagues [17] found that a higher BMI increases the risk of LEI in elite women football players. BMI is a direct indicator of the weight status of individuals [20]. In other words, a higher BMI refers to a higher inclination to be overweight or obese. As such, it is understandable that a higher BMI increases the risk of LEI in elite women football players. Another explanation of the high risk of injury could be related to the third law of physics where every action has an equal and opposite reaction. Since the body mass of high BMI players is high, the overall force, which is a product of the mass and acceleration with which they hit the ground, is also high, and this can cause a significantly greater impact on the lower extremities with more extensive injuries. Other contemporary studies have implied that overweight or obese players demonstrate lower control over their motions during games and are therefore more likely to have a higher incidence of injuries [21].

### 4.6. Limitations and Future Directions

Despite following a systematic database searching and screening method for data collection, the present systematic review and meta-analysis study are associated with certain limitations. Due to a lack of existing contemporary primary research on the topic of interest, only a limited amount of secondary data could be collected in this study. Due to this drawback, the findings of this study may not be generalizable for the global population of women football players [22]. One limitation of the study was the limited number of studies available for conducting a relevant meta-analysis, as the sample size is too small to compare the results on a more global level; hence, trials involving a larger sample size for more comparable results should be considered. Furthermore, due to time and resource constraints, only one researcher was involved in the screening of research articles in this study. This might have resulted in a certain degree of selection bias in this systematic review. Since the study had different predictors of LEI, the heterogeneity among the studies was high and even the baseline characteristics of all studies were not the same, increasing the risk of bias. However, the fact that only four relevant articles could be identified for this systematic review and meta-analysis study not only highlights a major limitation of this study, but also underscores an important knowledge gap pertaining to the domain of LEI in elite women football players.

Based on the limitations of the present study, as well as those identified from the existing body of evidence related to the topic of interest, the future direction of research related to the risk factors of LEI among elite women football players can be determined. Based on such an analysis, the following recommendations have been formulated:-Extensive primary research must be conducted to identify the risk factors of LEI among women football players, with a specific focus on elite women football players.-Experimental case–control research should be conducted to determine how addressing the factors identified in this study reduces the incidence of LEI among elite women football players.-Based on the findings of this study, the following recommendations have been formulated. These recommendations can be implemented to reduce the incidence and severity of LEI among elite women football players.-Fitness conditional coaches should focus on rectifying the postures and knee positions of players during matches.-Specialized injury prevention programmes should be designed and led by coaches for elite women football players.-Players should focus on reducing their BMI and increasing lumbopelvic control.-Trainers should study the history of LEI sustained by players in the past to devise strategic training programmes that will reduce the risk of future LEI in the players.-A questionnaire can be constructed, assessing all the relevant predictors, to provide relevant treatment beforehand.-Trials with larger sample sizes and similar baseline characteristics should be considered for limited heterogeneity.-The role of age and use of protective equipment needs to be explored.-The role of nutritional status in addition to BMI, and possible mechanisms involving the risk of BMI in LEI, needs further exploration.-Routine involvement of physiotherapists and occupational therapists can play a protective role.-A thorough screening and recovery time should be offered to players after each game.

## 5. Conclusions

This study provides a satisfactory list of predictors of LEI; however, more research is needed to establish their mechanisms of action. In total, eight factors that influence the risk of LEI among elite women football players were identified. Among these factors, six increase the risk of LEI, whereas two decrease the risk of LEI. Notably, the factors increasing the risk of LEI among elite women football players can be broadly divided into two categories: physical characteristics and history of injuries. Meanwhile, the two factors identified to decrease the risk of LEI among elite women football players are related to the practice and posture of the players. Based on the findings, it is concluded that the risk of LEI among the target population can be decreased by focusing on their physical characteristics, knee postures, and movements. However, due to a limited sample size and high heterogeneity, studies with homogenous results, similar baseline characteristics, and a larger sample size are required.

## Figures and Tables

**Figure 1 sports-11-00187-f001:**
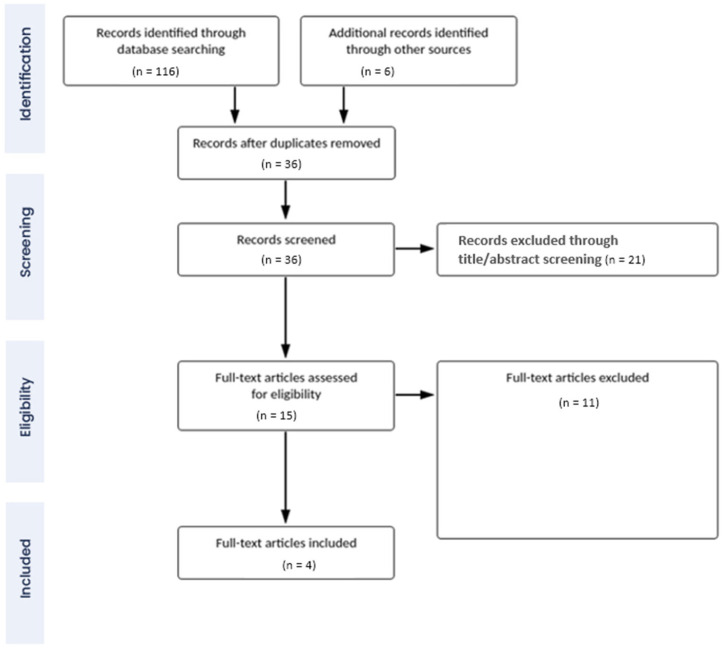
PRISMA flow diagram of study selection.

**Table 1 sports-11-00187-t001:** Eligibility criteria for research articles.

No.	Inclusion Criteria	Exclusion Criteria
1	The study specifically focused on the factors influencing the risk of lower extremity injuries in women football players.	The study did not focus on the factors influencing the risk of lower extremity injuries in women football players.
2	The study used an appropriate team of specialists to determine the level of injury.	The articles include children below the age of 13 years.
3	As the articles were published in 2013 or later, they are no more than ten years old.	The article focuses on other forms of injuries as a main consequence.
4	English is the language of publication for the article.	There is no English version of the article.

**Table 2 sports-11-00187-t002:** Format of the data extraction table.

Author(s)	Year of Publication	Sample Population	Number of Participants	Type and Prevalence of Lower Extremity Injury	Risk Factor Identified and Odds Ratio (OR)/Risk Ratio (RR)	CASP Score (Out of 10)

**Table 3 sports-11-00187-t003:** Data extraction table.

Author(s)	Type of Study	Sample Population	Number of Participants	Type; Prevalence of Lower Extremity Injury	Risk Factor Identified	CASP Score (Out of 10)	Notes
[17]	Cross-sectional cohort study	Elite female football players	173	All types of lower extremity injuries; 62%	Higher body mass index (BMI); lower knee valgus angle in a drop-jump landing; previous knee injury	9	
[18]	Prospective cohort study	Elite female football players from the German national league	143	All types of lower extremity injuries; 60.83%	Previous injury (anterior cruciate ligament rupture, ankle sprain, knee sprain)	9	Although the researchers aimed to identify the risk factors of injuries in elite female football players, all injuries included in the study were LEI.
[9]	Longitudinal cohort study	Female elite youth football players	351	All types of lower extremity injuries; 38.17%	Low normalized knee separation (≤10th percentile)	10	
[19]	Prospective cohort study	Female Swedish Premier League football players	228	All types of lower extremity injuries; 66%	Previous injury in the lower body	9	Although the researchers explored the risk factors of injuries in both men and women elite football players, the data related to only elite women football players were included in this meta-analysis

**Table 4 sports-11-00187-t004:** Findings.

**Author(s)**	**Number of Injuries**	**Risk Factor**	**Forest Plot** 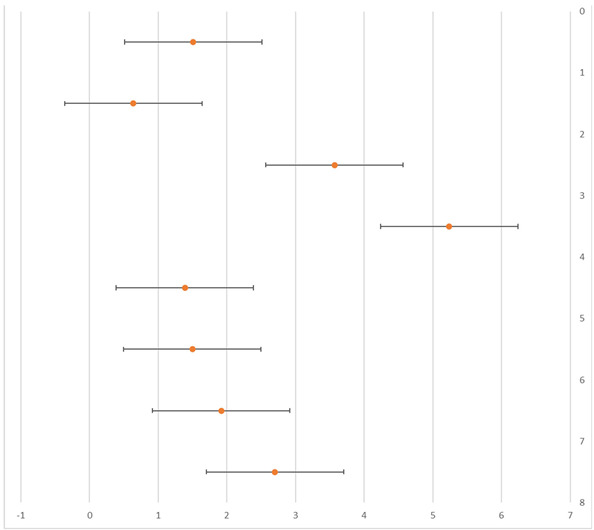	**OR/RR/HR (95%CI)**
[17]	171	Higher BMI		1.51 (1.08–2.11)
		Lower knee valgus angle in a drop-jump landing		0.64 (0.41–1.00)
		Previous knee injury		3.57 (1.27–9.99)
[18]	176	Previous ruptures anterior cruciate ligament rupture risk ankle sprain risk knee sprain risk		5.24 (1.42–19.59) 1.39 (0.62–3.10) 1.50 (0.61–3.72)
[9]	134	Low normalized knee separation during takeoff		1.92 (0.84–4.37)
[19]	1189	Previous injury in the lower body		2.7 (1.70–4.30)

**Table 5 sports-11-00187-t005:** Categorization of the identified factors.

Influence of the Factor	Category	Factors
Factors Increasing the Risk of LEI	Physical characteristics	High BMI Low normalized knee separation (≤10th percentile)
	Present characteristics and history	Previous knee injury History of LE injury Previous ruptures 1. Anterior cruciate ligament rupture risk; 2. Ankle sprain risk; 3. Knee sprain risk.
Factors Decreasing the Risk of LEI	Practice related	Lower knee valgus angle in a drop-jump landing

Abbreviations: LEI, lower extremity injuries.

## Data Availability

Data sharing is not applicable to this article.

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
