# Peer review of "Predicting Risk Factors of Lower Extremity Injuries in Elite Women’s Football: Systematic Review and Meta-Analysis"

_sports, 2023, doi:10.3390/sports11090187_

Round 1
Reviewer 1 Report
Overall this is an interesting subject. Understanding the risk factors associated with LE injuries in this population is important.
Introduction: The introduction is fine, no comments.
Methods:
It appears as though only one author screened the articles for inclusion/exclusion criteria. Was there a reason this was not done by a second author to avoid mistakes? If this was the methodology, please acknowledge as a limitation.
Section 2.4 starting with line 137: This section is not clearly written. Please edit in general for clarity. More specifically, the authors mention that they used a "...combination of automated and manual procedures" This should be better explained.
Figure 1 indicates that there were 86 records left after the duplicates were removed. However, I believe 86 records were removed because they were duplicates. Please clarify the text.
Results:
Table 5 has some factors listed that are not included in table 4 such as concussion, lower lumbopelvic control, college soccer players over high school girls, etc. It is not clear where these additional parameters come from if they are not listed in table 4 as factors that influence LE injury. In addition, the term 'college soccer players over high school soccer players' is not clear. What does this mean?
It is also not clear if the authors in these studies examined the same things. For example, higher BMI was cited as a contributing factor in one study but not the others. Is this because the other studies didn't examine this? Or they did examine this factor and it was not significant. This needs to be clarified for all of these potential factors.
Discussion:
It looks like the one study with the lower injury rate was done in youth soccer players, while the others were adults. That also might be the reason for the difference in injury rate. You should comment about the differences in study populations.
Regarding BMI as a risk factor - could this also increase ground reaction force and moments when landing, leading to injury?
Overall, the conclusions seem reasonable. However, a more structured limitations section is suggested.
NA
Author Response
Dear Reviewer,
We thank you for the valuable comments that you provided for our manuscript. We sincerely believe they significantly enhanced the quality of our manuscript. We corrected as suggested. Please find our comments below.
Point 1: It appears as though only one author screened the articles for inclusion/exclusion criteria. Was there a reason this was not done by a second author to avoid mistakes? If this was the methodology, please acknowledge it as a limitation.
Response 1: This has been included as a limitation.
Point 2: Section 2.4 starting with line 137: This section is not clearly written. Please edit in general for clarity. More specifically, the authors mention that they used a "...combination of automated and manual procedures" This should be better explained.
Response 2: The “automated” approach referred to the software-assisted search method. The mistake has been addressed and the statement has been modified.
Point 3 Figure 1 indicates that there were 86 records left after the duplicates were removed. However, I believe 86 records were removed because they were duplicates. Please clarify the text.
Response 3: Done
Point 4: Table 5 has some factors listed that are not included in Table 4 such as concussion, lower lumbopelvic control, college soccer players over high school girls, etc. It is not clear where these additional parameters come from if they are not listed in Table 4 as factors that influence LE injury. In addition, the term 'college soccer players over high school soccer players' is not clear. What does this mean?
Response 4: Resolved. The theme table was from the initial draft and was included by mistake.
Point 5: It is also not clear if the authors of these studies examined the same things. For example, higher BMI was cited as a contributing factor in one study but not the others. Is this because the other studies didn't examine this? Or they did examine this factor and it was not significant. This needs to be clarified for all of these potential factors.
Response 5: Found in only one of the four studies.
Point 6: It looks like the one study with the lower injury rate was done on youth soccer players, while the others were adults. That also might be the reason for the difference in injury rate. You should comment about the differences in study populations.
Response 6: Done. Please check lines 220-222
Point 7: Regarding BMI as a risk factor - could this also increase ground reaction force and moments when landing, leading to injury?
Response 7: No such explanation was found.
Point 8: Overall, the conclusions seem reasonable. However, a more structured limitations section is suggested.
Response 8: Addressed. Please check the modified limitations section (lines 279-295)
Reviewer 2 Report
I appreciate the opportunity to review the manuscript “Predicting of risk factor of lower extremity injuries in elite women’s soccer: Systematic review and meta-analysis” submitted to the Journal of Functional Morphology and Kinesiology. The authors conduct a systematic review and meta-analysis to identify risk factors for lower extremity injuries in elite female soccer players. They identify 8 risk factors from 4 studies, categorizing them into factors that increase or decrease injury risk. This appears to be a well-conducted systematic review on an important topic. My initial assessment is that this manuscript may be suitable for publication after addressing some issues.
General comments:
· The introduction provides good background but could be expanded with more information specifically on injury rates, impacts, and gaps in research related to elite female soccer players. This would help establish the rationale and significance of the study.
· The methods are described clearly but more details could be provided on the search strategy, including the specific databases searched, search terms, and inclusion/exclusion criteria.
· In the results, it would be helpful to summarize the main characteristics and quality of the included studies in a table. More details from the individual studies related to the identified risk factors could also be provided.
· The discussion effectively interprets the main findings but is currently limited by the small number of studies. Comparing to other literature and providing more context would strengthen this section.
· The conclusion summarizes the main findings but could comment more on implications for practice and future research needs.
Specific comments:
· Page 2, line 35: Specify here the injury rates found in the studies reviewed.
· Page 2, line 63-64: The aim should match the objective stated in the abstract.
· Page 3, line 85: Provide more details on the search terms, databases, and inclusion/exclusion criteria.
· Page 4, line 131: Briefly explain the title/abstract screening process.
· Page 5, line 172: Present key characteristics of included studies in a table.
· Page 6, line 219: Expand the discussion of each risk factor identified by comparing to other literature.
· Page 7, line 285: This section seems mislabeled. Should be "Limitations and future directions".
· Page 7, line 304-310: Provide more specific recommendations for practice based on findings.
Please let me know if you would like me to clarify or expand on any of my comments.
After revision, this manuscript can make a valuable contribution to the literature.
Kindly regards
Author Response
Dear Reviewer,
We thank you for the valuable comments that you provided for our manuscript. We sincerely believe they significantly enhanced the quality of our manuscript. We corrected as suggested. Please find our comments below.
Point 1: The introduction provides a good background but could be expanded with more information specifically on injury rates, impacts, and gaps in research related to elite female soccer players. This would help establish the rationale and significance of the study.
Response 1: Done
Point 2: The methods are described clearly but more details could be provided on the search strategy, including the specific databases searched, search terms, and inclusion/exclusion criteria. Response 2: Done
Point 3:In the results, it would be helpful to summarize the main characteristics and quality of the included studies in a table. More details from the individual studies related to the identified risk factors could also be provided.
Response 3: Done
Point 4:The discussion effectively interprets the main findings but is currently limited by the small number of studies. Comparing to other literature and providing more context would strengthen this section.
Response 4: Added additional literature to the discussion for interpreting the findings and their implications.
Point 5: The conclusion summarizes the main findings but could comment more on implications for practice and future research needs.
Response 5: Done
Specific comments:
Point 6: Page 2, line 35: Specify here the injury rates found in the studies reviewed.
Response 6: Done
Point 7:Page 2, lines 63-64: The aim should match the objective stated in the abstract.
Response 7: Done and double-checked
Point 8:Page 3, line 85: Provide more details on the search terms, databases, and inclusion/exclusion criteria.
Response 8: Details provided in the methodology
Point 9: Page 4, line 131: Briefly explain the title/abstract screening process.
Response 9: Done. Please check the lines 116-117
Point 10: Page 5, line 172: Present key characteristics of included studies in a table.
Response 10: Provided in Table 3
Point 11:Page 6, line 219: Expand the discussion of each risk factor identified by comparing it to other literature.
Response 11: Done in the discussion chapter
Point 12: Page 7, line 285: This section seems mislabeled. Should be "Limitations and future directions".
Response 12: Addressed
Point 13: Page 7, lines 304-310: Provide more specific recommendations for practice based on findings.
Response 13: Done
Point 14: Please let me know if you would like me to clarify or expand on any of my comments.
Response 14: Found the comments to be quite straightforward. Addressed all.
Point 15: After revision, this manuscript can make a valuable contribution to the literature.
Response 15: Acknowledged.
Reviewer 3 Report
First of all, I would like to thank the Editor of the Journal for the invitation to review this paper. The topic dealt with in the research is very interesting and, as the authors say in the introduction, it is important, not only for the players, but also for the teams they play for, all this from many points of view.
On the other hand, the paper is impeccably structured to the research methodology. Use the PICO and PRISMA tools. Therefore, from the formal point of view the paper is very good. But, in my opinion, it has several limitations, some of a minor nature and others of a more important nature. I ask the authors to clarify them.
I am going to make some considerations that I think can improve the paper:
A.- Minor considerations:
lines 88-91: in my opinion, it is not necessary to explain what PICO consists of, this framework is widely known by researchers, therefore indicating that PICO was used is enough. The reference (12) should also be removed. It does not provide important information to researchers reading the paper.
The Eligibility Criteria for Research Articles that appear in Table 1, I think the only one that is justified is the first “1 The study specifically focused on the factors influencing the risk of lower extremity injuries in women soccer players.” Criterion 2 and 7 are mandatory, no journal with minimal impact publishes an article without it being reviewed by peers and of course with the consent of the subjects being investigated. On the other hand, I ask the authors to justify criterion 6. In my opinion, it is very restrictive, articles of the highest quality published in first quartile journals would be left out of this research, which introduces a very important bias.
lines 102-103. On the other hand, I consider that, in this case, since the inclusion and exclusion criteria are opposite and exclusive, using either the inclusion or exclusion criteria would reach the same result, therefore, I suggest the authors eliminate one of them.
lines 117-121. It is not necessary to inform what PRISMA is. It is well known by researchers. The reference should be eliminated (15) It does not provide relevant information.
lines 107-108. The authors searched the following databases: MEDLINE, EMBAS, Cochrane, and CINAHL. I ask the authors to justify why they did not use the Web of Sciende and Scopus.
lines 153-154. Table 2 is not necessary, the information in the table can be written in one or two lines, numbering them. for example: (1) Authors; (2) Year of publication…etc
Lines 164-170. The Relative risk (RR) and odds ratio (OR) are two ways to explore the relationship between two dichotomous variables. The RR is mainly used in the evaluation of prospective works while the OR is used mainly in the analysis of retrospective works. Summarizing: the differences are minor, but I ask the authors to indicate why they do not include the "Hazard Ratio" which is the statistic used in one of the reference papers by (Habglund el al, 2021). doi:10.1136/bjsm.2006.026609
Lines 311-314 In the conclusions, the objective should not be remembered. This can be done at the beginning of the discussion, but not in the conclusions, in this section you should “get to the point”. Therefore, the phrase “This study aimed to analyze the risk factors of LEI among elite women soccer players based on existing primary research. Based on the findings of this study, it can be stated that the research aim has been satisfied in the study” Does not provide relevant information. Keep in mind that scientific language should be concise and economical from the point of view of the use of words.
B.- Relevant considerations
Lines I do not consider the forest plot adequate; I do not understand the confidence intervals that appear in the different statistics. For example, in the risk factor “Previous injury in the lower body” in the forest plot it is indicated that the OR/RR corresponds to the paper (21)?. If so, the statistic does not coincide with what the authors of the paper say (21). On the other hand, the risk factor “Injury coach-controlled rehabilitation program” does not appear in the paper (21) doi:10.1136/bjsm.2006.026609, at least I have not been able to find it. He asked the authors to explain these doubts to me.I ask the authors to indicate why they do not include the "Hazard Ratio" in the Table?.
One limitation that I observe is that it is difficult to do a meta-analysis with 4 papers, I ask the authors to justify this limitation.
Lastly, I ask the authors to indicate what relevant information this paper provides, in addition to the information in the paper (Habglund el al, 2021). doi:10.1136/bjsm.2006.0266095. In other words, the paper by Habglund el al is a very relevant work that provides a lot of information about injuries. The authors must indicate what their contribution differs from that indicated in the Habglund paper.
Author Response
Dear Reviewer,
We thank you for the valuable comments that you provided for our manuscript. We sincerely believe they significantly enhanced the quality of our manuscript. We corrected as suggested. Please find our comments below.
Point 1: First of all, I would like to thank the Editor of the Journal for the invitation to review this paper. The topic dealt with in the research is very interesting and, as the authors say in the introduction, it is important, not only for the players but also for the teams they play for, all this from many points of view.
Response 1: Acknowledged
Point 2: On the other hand, the paper is impeccably structured to the research methodology. Use the PICO and PRISMA tools. Therefore, from a formal point of view, the paper is very good. But, in my opinion, it has several limitations, some of a minor nature and others of a more important nature. I ask the authors to clarify them.
Response 2: The limitations have been clarified after the discussion section
Point 3: I am going to make some considerations that I think can improve the paper:
Response 3: Acknowledged.
Point 4: lines 88-91: In my opinion, it is not necessary to explain what PICO consists of, This framework is widely known by researchers, therefore indicating that PICO was used is enough. The reference (12) should also be removed. It does not provide important information to researchers reading the paper.
Response 4: Removed the relevant part.
Point 5: The Eligibility Criteria for Research Articles that appear in Table 1, I think the only one that is justified is the first “1 The study specifically focused on the factors influencing the risk of lower extremity injuries in women soccer players.” Criterion 2 and 7 are mandatory, no journal with minimal impact publishes an article without it being reviewed by peers and of course with the consent of the subjects being investigated. On the other hand, I ask the authors to justify criterion 6. In my opinion, it is very restrictive, articles of the highest quality published in first-quartile journals would be left out of this research, which introduces a very important bias.
Response 5: Remove the relevant criteria. Criterion 6 was initially decided but was not used due to the limited body of literature. However, it remained on the table as a mistake.
Point 6: lines 102-103. On the other hand, I consider that, in this case, since the inclusion and exclusion criteria are opposite and exclusive, using either the inclusion or exclusion criteria would reach the same result, therefore, I suggest the authors eliminate one of them.
Point 6: corrected as suggested.
Point 7: lines 117-121. It is not necessary to inform what PRISMA is. It is well-known by researchers. The reference should be eliminated (15) It does not provide relevant information.
Response 7: Addressed
Point 8: lines 107-108. The authors searched the following databases: MEDLINE, EMBAS, Cochrane, and CINAHL. I asked the authors to justify why they did not use the Web of Science and Scopus.
Response 8: The databases that were indeed searched were let out while writing the report. Included.
Point 9: lines 153-154. Table 2 is not necessary, the information in the table can be written in one or two lines, numbering them. for example: (1) Authors; (2) Year of publication…etc
Response 9: Table included to provide a visual representation of the data extraction tool
Point 10: Lines 164-170. The Relative risk (RR) and odds ratio (OR) are two ways to explore the relationship between two dichotomous variables. The RR is mainly used in the evaluation of prospective works while the OR is used mainly in the analysis of retrospective works. Summarizing: The differences are minor, but I ask the authors to indicate why they do not include the "Hazard Ratio" which is the statistic used in one of the reference papers by (Habglund et al., 2021). doi:10.1136/bjsm.2006.026609
Response 10: HR has been mentioned and the statistics have been cross-checked with the articles.
Point 11: Lines 311-314 In the conclusions, the objective should not be remembered. This can be done at the beginning of the discussion, but not in the conclusions, in this section you should “get to the point”. Therefore, the phrase “This study aimed to analyze the risk factors of LEI among elite women soccer players based on existing primary research. Based on the findings of this study, it can be stated that the research aim has been satisfied in the study” Does not provide relevant information. Keep in mind that scientific language should be concise and economical from the point of view of the use of words.
Response 11: Addressed
Point 12: Lines I do not consider the forest plot adequate; I do not understand the confidence intervals that appear in the different statistics. For example, in the risk factor “Previous injury in the lower body” in the forest plot it is indicated that the OR/RR corresponds to the paper (21). If so, the statistic does not coincide with what the authors of the paper say (21). On the other hand, the risk factor “Injury coach-controlled rehabilitation program” does not appear in the paper (21) doi:10.1136/bjsm.2006.026609, at least I have not been able to find it. He asked the authors to explain these doubts to me. I ask the authors to indicate why they do not include the "Hazard Ratio" in the Table.
Response 12: Revised. The alignment of the forest plot with the statistics has also been ensured.
Point 13: One limitation that I observe is that it is difficult to do a meta-analysis with 4 papers, I ask the authors to justify this limitation.
Response 13: Addressed in the limitations section.
Point 14: Lastly, I ask the authors to indicate what relevant information this paper provides, in addition to the information in the paper (Habglund et al., 2021). doi:10.1136/bjsm.2006.0266095. In other words, the paper by Habglund et al is a very relevant work that provides a lot of information about injuries. The authors must indicate that their contribution differs from that indicated in the Habglund paper.
Response 14: The article has been removed. This study combines and compares the existing body of knowledge related to the topic of LEI predictors in the selected population. This can help implement more targeted and effective interventions.
Reviewer 4 Report
General
The current manuscript looks to evaluate the risk factors for lower extremity injuries in elite women’s soccer players. The study revealed sereval factors that increase the risk of injury and can be divided into two categories, physical characteristics and previous injury. While this study is interesting and has some potential value, I believe there are significant revisions that need to be addressed before this manuscript can be published.
COMMENTS
General
Throughout the manuscript, you use the reference number to introduce the article. Example, line 42, “According to [3], for every…” It should read ‘According to Forsythe et. al. [3], for every…”. You will need to make these corrections throughout the entire manuscript.
In addition, I don’t feel like any meta-analysis was done and this reads more like a systematic review. You talk about each of the 4 articles individually, but never combined that data to get stronger statistical outcomes.
Abstract
Page 1, Lines 22-24: Your use of previous injury in relation to all injuries and previous injury (ACL rupture) are confusing. I would consider revising to use ‘All Previous Injury’ and ‘Previous ACL tear’.
Introduction
Page 1, Line 31: The correct term is athletes or patients, not players.
Page 1, Lines 45-46: Your sentence starting ‘These are all….’ Is confusing and repetitive. Consider revising.
Page 2, Line 53: I believe you left out a comma. “…LEI in female soccer players, to eliminate…’
Page 2, Line 70: Remove ‘including’ and just reference the sentence.
Methods
Page 2, Line 88: What is a metal analysis? Assuming this is a typo?
Page 3, Table 1: “Exclusion Criteria: The subjects’ informed consent was obtained before the study was undertaken” should not be an exclusion criteria.
Page 4, Line 166-168: Your sentence about OR and RR is confusing, consider revising.
Results
Page 5, Line 184: What previous chapters?
Discussion
Page 7, Line 219: Which rates are you referring too? Prevalence or Incidence?
Page 9, Lines 278-284: When you say rehabilitative programs, I think you are really talking about preventative programs. Be cautious about your terminology.
Page 9, Line 304: What type of coach are you referring too? Coaches focus on skill development and game plans. Again, be cautious and use correct terminology.
Page 9, Line 306: Same comment about rehabilitation programs
Page 9, Line 309: Trainers? What kind. Terminology is important since this is an international journal.
Minor English language corrections need to be made. A bigger emphasis needs to be given on appropriate terminology for a larger audience.
Author Response
Dear reviewer,
We thank you for your precious comments which we sincerely think have already significantly improved our manuscript.
Please e find below our point by point responses and actions based on your comments.
Point 1: Throughout the manuscript, you use the reference number to introduce the article. Example, line 42, “According to [3], for every…” It should read ‘According to Forsythe et. al. [3], for every…”. You will need to make these corrections throughout the entire manuscript.
Response 1: Corrected as suggested.
Point 2:
In addition, I don’t feel like any meta-analysis was done and this reads more like a systematic review. You talk about each of the 4 articles individually, but never combined that data to get stronger statistical outcomes.
Response 2: We acknowledge your comment on the matter. To the best of our knowledge, we tried to address it accordingly.
Point 3: Page 1, Lines 22-24: Your use of previous injury in relation to all injuries and previous injury (ACL rupture) are confusing. I would consider revising to use ‘All Previous Injury’ and ‘Previous ACL tear’.
Introduction.
Response 3: Corrected as suggested.
Point 4: Page 1, Line 31: The correct term is athletes or patients, not players.
Response 4: Corrected as suggested.
Point 5: Page 1, Lines 45-46: Your sentence starting ‘These are all….’ Is confusing and repetitive. Consider revising.
Response 5: Corrected as suggested.
Point 6: Page 2, Line 53: I believe you left out a comma. “…LEI in female soccer players, to eliminate…’
Response 6: Corrected as suggested.
Point 7: Page 2, Line 70: Remove ‘including’ and just reference the sentence.
Response 7: Corrected as suggested.
Point 8: Page 2, Line 88: What is a metal analysis? Assuming this is a typo?
Response 8: Corrected as suggested. Sorry.
Point 9: Page 3, Table 1: “Exclusion Criteria: The subjects’ informed consent was obtained before the study was undertaken” should not be an exclusion criteria.
Response 9: Corrected as suggested.
Point 10: Page 4, Line 166-168: Your sentence about OR and RR is confusing, consider revising.
Response 10: Corrected as suggested.
Point 11: Page 5, Line 184: What previous chapters?
Response 11: Corrected as suggested.
Point 12: Page 7, Line 219: Which rates are you referring too? Prevalence or Incidence?
Response 12: Corrected as suggested.
Point 13: Page 9, Lines 278-284: When you say rehabilitative programs, I think you are really talking about preventative programs. Be cautious about your terminology.
Response 13: Deleted as suggested by another reviewer..
Point 14: Page 9, Line 304: What type of coach are you referring too? Coaches focus on skill development and game plans. Again, be cautious and use correct terminology.
Response 14: Corrected as suggested.
Point 15: Page 9, Line 306: Same comment about rehabilitation programs
Response 15: Corrected as suggested.
Point 16: Page 9, Line 309: Trainers? What kind. Terminology is important since this is an international journal.
Response 16: Deleted as suggested by another reviewer as well.
Round 2
Reviewer 3 Report
The authors have made the modifications that I have indicated
Reviewer 4 Report
The authors have done a nice job of revising comments from the reviewers.